# Citicoline/Coenzyme Q10/Vitamin B3 Fixed Combination Exerts Synergistic Protective Effects on Neuronal Cells Exposed to Oxidative Stress

**DOI:** 10.3390/nu14142963

**Published:** 2022-07-20

**Authors:** Leonardo Mastropasqua, Luca Agnifili, Claudio Ferrante, Matteo Sacchi, Michele Figus, Gemma Caterina Maria Rossi, Lorenza Brescia, Raffaella Aloia, Giustino Orlando

**Affiliations:** 1Ophthalmology Clinic, Department of Medicine and Aging Science, University G. d’Annunzio of Chieti-Pescara, 66100 Chieti, Italy; mastropa@unich.it (L.M.); raffaellaaloia77@gmail.com (R.A.); 2Department of Pharmacy, G. d’Annunzio University, 66013 Chieti, Italy; claudio.ferrante@unich.it (C.F.); giustino.orlando@unich.it (G.O.); 3University Eye Clinic, San Giuseppe Hospital, IRCCS Multimedica, 20900 Milan, Italy; matteosacchi.hsg@gmail.com; 4Ophthalmology Unit, Department of Surgery, Medicine, Molecular and Emergency, University of Pisa, 56124 Pisa, Italy; figusmichele@gmail.com; 5Fondazione IRCCS Policlinico San Matteo, 27100 Pavia, Italy; g.rossi@smatteo.pv.it

**Keywords:** neurodegenerative diseases, citicoline, niacin, coenzyme Q10, associative therapy, neuroprotection, neuroinflammation, oxidative stress

## Abstract

Background: The present study aimed to investigate the rationale and efficacy of using a citicoline, coenzyme Q10 (CAVAQ10) and vitamin B3 fixed combination in combating inflammation and oxidation in neuronal cells exposed to oxidative stress. Methods: HypoE22 cells and isolated hypothalamic specimens were selected as in vitro models to conduct the experiments. The efficacy of citicoline, CAVAQ10, and vitamin B3, with their fixed combination, were assayed after the exposure of hypothalamic cells to hydrogen peroxide (concentration range 1 nM–10 µM), in order to evaluate the biocompatibility of treatments. The activity of neuroprotective and pro-inflammatory factors, namely, brain-derived neurotrophic factor (BDNF), interleukin-6 (IL-6), and tumor necrosis factor-α (TNFα), involved in the neuronal cell damage in neurodegenerative diseases, were assayed in isolated hypothalamus. Results: Neither citicoline, CAVAQ10, nor vitamin B3 significantly altered hypothalamic cell viability, thus suggesting the biocompatibility of single ingredients and fixed combination in the concentration range considered for the study. In the same condition, citicoline and CAVAQ10 were also effective in reducing the gene expression of monoaminoxidase-B, involved in dopamine degradation. However, only citicoline demonstrated an ability to reduce dopamine levels. Conversely, all compounds were effective in reducing the gene expression of IL-6, and TNFα, and in inducing the gene expression of BDNF, with the co-administration of citicoline/CAVAQ10/vitamin B3 being generally more effective than single ingredients. Conclusions: The present findings support the beneficial and synergistic effects of citicoline, CAVAQ10, and vitamin B3 in fixed combination in reducing inflammation and oxidation, and in stimulating neurotrophin production in neuronal cells.

## 1. Introduction

Neurodegenerative diseases (NDs) occur when neuronal cells progressively lose structure and function over time. Their prevalence increases with age and, because of disabilities they produce, they have a huge socio-economic burden [1]. Unfortunately, dedicated treatments are not available for most NDs, with aspecific neuroprotective approaches being the only available strategies aimed at containing the neural damage progression [2]. These approaches generally aim to protect, recover, or regenerate the structure and function of neuronal cells. Among the countless molecules exerting neuroprotective effects, citicoline and coenzyme Q10 (Q10) are among the most investigated and used in clinical practice for some forms of ND [3,4,5,6]. Citicoline stimulates the biosynthesis of phospholipids and their precursors, such as phosphatidylcholine, which are essential for the proper maintenance of the cellular membrane architecture. Since choline participates in the biosynthesis of neurotransmitters (acetylcholine), citicoline is considered a molecule with structural and functional activity [5]. The neuromodulatory and neuroprotective properties of citicoline were extensively investigated either in vivo or in vitro [6,7]. As a neuromodulator, citicoline was shown to increase, in in vivo studies, the levels of neurotransmitters (dopamine, noradrenaline, and serotonin) in the central nervous system [8]. On the other hand, its neuroprotective properties were documented in in vitro models of neuroblastoma cells, dopaminergic mesencephalic neurons, on cultured retinal neurons, and in clinical studies on glaucoma and diabetic retinopathy [9,10,11,12,13,14,15]. Coenzyme Q 10 (Q10) is an endogenous lipophilic quinone, which exerts its functions within mitochondria, where it acts as an electron accepter, limits oxidative stress and inflammation, and prevents apoptosis [16,17]. Several studies supported the neuroprotective effects of Q10 in degenerative disorders such as Alzheimer’s, Huntington’s, and Parkinson’s diseases [18,19]. In addition, Q10 had neuroprotective effects in experimental models of ocular NDs, such as glaucoma, retinitis pigmentosa, and retinal vascular disease [6,17,18,20,21]. Based on evidence that a diet supplemented with nicotinamide protects retinal ganglion cells (RGCs) from degeneration and prevents glaucoma in aged mice, the scientific community began to include vitamin B3 (nicotinamide, niacin) in neuroprotective strategies [22]. Recently, it was observed that nicotinamide levels were reduced in patients with glaucoma, and that vitamin B3 supplementation improved inner retinal function [23,24]. As observed in many forms of NDs, the mechanisms underlying the neuronal damage were often multifactorial [25,26]. Thus, the possibility of identifying neuroprotective molecules, acting at different levels in the cascade of events leading to neuronal cell loss, is a step forward in resolving a strong clinical need and for expanding an emerging field of research [17,25].

The aims of the present study were to use an oxidative stress model of injury to investigate: (i) the effects of the fixed combination of citicoline/coenzyme Q10/vitamin B3 on neuronal cell (hypothalamic HypoE22 cell line) viability; (ii) whether this association presents antioxidant effects on damaged neuronal cells, through the evaluation of the brain-derived neurotrophic factor (BDNF) gene expression; and (iii) whether the fixed combination of citicoline/niacin/Q10 has synergistic effects over each single component. The HypoE22 cell was chosen as the neuronal in vitro model for investigating the biocompatibility limits of the fixed combination citicoline/coenzyme Q10/vitamin B3 [27,28]. We also investigated the role of citicoline, vitamin B3, coenzyme Q10, and their pharmacological impact on the gene expression of brain-derived neurotrophic factor (BDNF), interleukin-6, and tumor necrosis factor-α (TNFα) in isolated hypothalami. An in silico study was conducted with the aim of unravelling the mechanisms of action underlying observed effects.

## 2. Materials and Methods

### 2.1. Bioactive Components of the Food Supplements

The bioactive components of the food supplements citicoline (cytidine diphosphate choline sodium, 567.2 mg), vitamin B3 and coenzyme Q10 (CAVAQ10^®^, 200 mg) were provided as dried materials by OMEGA PHARMA Srl (Cantù, Italy). CAVAQ10^®^, an inclusion complex made of natural ubiquinone with CAVAMAX^®^ W8 gamma-cyclodextrin, is a highly bioavailable coenzyme Q10 powder with improved dispersibility in water.

### 2.2. Hypothalamic HypoE22 Cells: Evaluation of Biocompatibility

Hypothalamic HypoE22 cells were purchased from Cedarlane Cellution Biosystem and cultured in DMEM (Euroclone) supplemented with 10% (*v*/*v*) heat-inactivated fetal bovine serum and 1.2% (*v*/*v*) penicillin G/streptomycin in a 75 cm^2^ tissue culture flask (*n* = 5 individual culture flasks for each condition). The culture conditions and the viability 3-(4,5-dimethylthiazol-2-yl)-2,5-diphenyltetrazolium bromide (MTT) test were performed as previously described [27,28]. The effects of the treatment (24 h) on cell viability were evaluated in comparison to the untreated control group.

### 2.3. Isolated Hypothalamic Specimens Exposed to Oxidative Stress

Twenty-four male adult mice C57BL6 were housed in Plexiglass cages (40 cm × 25 cm × 15 cm), two mice per cage, in climatized colony rooms (22 ± 1 °C; 60% humidity), on a 12 h/12 h light/dark cycle (light phase: 07:00–19:00 h). They had free access to tap water and food, 24 h/day throughout the study, with no fasting periods. Mice were fed a standard laboratory diet (3.5% fat, 63% carbohydrate, 14% protein, 19.5% other components without caloric value, 3.20 kcal/g). Housing conditions and experimentation procedures were strictly in accordance with the European Union ethical regulations on the care of animals for scientific research. The experimental paradigm was approved by the Local Ethical Committee (University “G. d’Annunzio” of Chieti-Pescara) and the Italian Health Ministry (Project no. 885/2018-PR). Specifically, mice were sacrificed by CO_2_ inhalation (100% CO_2_ at a flow rate of 20% of the chamber volume per min) and hypothalami were immediately collected and maintained in a humidified incubator with 5% CO_2_ at 37 °C for 4 h (incubation period), in DMEM enriched with the treatment and exposed to hydrogen peroxide 500 µM. Afterwards, the hypothalamic samples were subjected to analytical procedure for gene expression analysis.

### 2.4. Gene Expression Analysis

The gene expression of BDNF, TNFα, and IL-6 was conducted as previously reported [28]. Briefly, after extraction through the TRI Reagent, total RNA was reverse-transcribed using a High-Capacity cDNA Reverse Transcription Kit (ThermoFischer Scientific, Waltman, MA, USA). Gene expression was determined by quantitative real-time PCR using TaqMan probes, and β-actin was used as the housekeeping gene. Data analysis was carried out with the Sequence Detection System (SDS) software version 2.3 (ThermoFischer Scientific, Waltman, MA, USA).

### 2.5. In Silico

The prediction of the putative mechanism of action of citicoline, vitamin B3 and CAVAQ10 was carried out through the platform STITCH.

### 2.6. Statistical Analysis

The experimental data related to in vitro and ex vivo studies were analyzed through the analysis of variance (ANOVA) followed by Newman–Keuls post hoc test. The GraphPad Prism software was employed for statistical analysis. *p* < 0.05 was considered statistically significant. 

## 3. Results

Figure 1 shows the null effect of citicoline, vitamin B3, and CAVAQ10^®^ (1 nM–10 µM) on hypothalamic HypoE22 cell viability. Basically, the fixed combination did not significantly alter the hypothalamic cell viability. This indicated biocompatibility, in the abovementioned concentration range, which was also selected for the subsequent ex vivo tests on isolated hypothalamic tissues.

In this context, it is important to note that the citicoline/vitamin B3/CAVAQ10^®^ (10 µM) association was the most effective in stimulating the gene expression of BDNF and reducing that of TNFα (Figure 2 and Figure 3). On the other hand, there were no significant differences between the blunting effect on IL-6 gene expression induced by citicoline/vitamin B3/CAVAQ10^®^ (10 µM), compared with single ingredients. Nevertheless, the levels of IL-6 mRNA were also lower compared to the untreated control group. This could suggest the ability of the tested formula in counteracting and/or partially preventing the burdens of inflammation and oxidative stress that occur in chronic inflammatory disorders, including glaucoma.

The results of the in silico study, conducted on the platform STITCH, are summarized in Figure 3. Specifically, citicoline was predicted to interact with monoamine oxidase A (MAO-A), dopamine carrier (SLC22A2), and phosphate cytidyltransferases (PCYT1B and PCYT1A). CAVAQ10 was predicted to interact with cytochrome b (MT-CYB) and fibroblast growth factor-2 (FGF2). Finally, putative interactions were predicted between vitamin B3 and proteins involved in nicotinate and nicotinamide metabolism, among which are purine nucleoside phosphorylase (PNP) and nicotinate phosphoribosyl transferase domain containing 1 (NAPRT1).

## 4. Discussion

To date, citicoline and coenzyme Q10 represent some of the most promising neuroprotective agents available in ophthalmology, with growing evidence of their potential utility in NDs, especially glaucoma [6,14,15,16,29,30,31,32]. Along with these two compounds, recent findings highlighted the potential utility of vitamin B3 in the protection of retinal ganglion cells [22].

In the present in vitro study, citicoline, vitamin B3, and CAVAQ10^®^ were assayed to investigate their safety and efficacy in neuronal cell protection and determine whether their fixed combination (now commercialized in Italy under the name of Retigan Q10^®^, OMEGA PHARMA Srl (Cantù, Italy)) was more effective in reducing neuroinflammation and in favoring cellular metabolism compared to each single compound. The hypothalamic HypoE22 cell line was chosen according to previous papers to predict the biocompatibility of each single ingredient [28]. Moreover, as previously reported, hypothalamic cells can be efficiently used to explore neuroprotective effects of citicoline and coenzyme Q10 [7,33].

A viability MTT test was conducted in basal conditions, and the results highlighted the complete biocompatibility of citicoline, vitamin B3, and CAVAQ10^®^ in the concentration range of 1 nM–10 µM. Additionally, when the compounds were co-administered at the highest concentration tolerated by the cells (10 µM), in basal conditions, the biocompatibility was preserved, and the cell viability, in all pharmacological treatments, was always >70% compared to the untreated cells. 

In line with these results, citicoline, vitamin B3 and CAVAQ10^®^ were also effective in reducing the burden of oxidative stress in isolated hypothalamic specimens. To induce the neuron oxidative stress, hydrogen peroxide 300 µM was chosen as pro-oxidant stimulus [27]. In this context, the fixed combination citicoline, vitamin B3 and CAVAQ10^®^ was the most effective in blunting the hydrogen peroxide-induced downregulation of BDNF gene expression. This aspect is noteworthy, since BDNF was reported to be involved in the pathogenesis of some major neurodegenerative diseases of the eye, such as glaucoma [34,35,36,37,38]. The blunting effects induced by citicoline, vitamin B3 and CAVAQ10^®^ on BDNF downregulation further suggested the involvement of this neurotrophin in the neuroprotection induced by both compounds [37,38]. Furthermore, the protective effects of citicoline, vitamin B3 and CAVAQ10^®^ were also corroborated by their efficacy in inhibiting the hydrogen peroxide-induced up-regulation of both IL-6 and TNFα. These results indicated an important anti-neuroinflammatory effect of this association, which seems to be crucial to contain the neuronal loss in some NDs [39]. This could also suggest the ability of the tested formula in limiting and/or partially preventing the burdens of inflammation and oxidative stress that occur in chronic inflammatory disorders, including glaucoma.

The fixed combination was more effective than single ingredients, confirming the rationale for their association to contain the burdens of oxidation and inflammation.

The results of the components-targets analysis (Figure 3) conducted via the platform STITCH suggested the capability of citicoline to interact with MAO-A enzymes, which could suggest potential influences on dopamine turnover [40,41]. Additionally, citicoline was predicted to modulate dopamine uptake, through the interaction with the neurotransmitter carrier (SLC22A2), and via phosphatidylcholine synthesis, where it interacts with phosphate cytidylyltransferases (PCYT1B and PCYT1A). On the other hand, CAVAQ10 was predicted to interact with cytochrome b (MT-CYB), involved in the ATP biosynthesis, and fibroblast growth factor-2 (FGF2). In addition, data from the literature pointed to the inhibition of FGF2-induced angiogenesis by CAVAQ10 [42]. Finally, vitamin B3 was predicted to interact with numerous proteins involved in nicotinate and nicotinamide metabolism, among which were purine nucleoside phosphorylase (PNP) and nicotinate phosphoribosyl-transferase domain containing 1 (NAPRT1). Particularly, nicotinamide has been suggested to act as neuroprotective agent in open-angle glaucoma [22,43,44]. Collectively, the present in silico data suggest that citicoline, vitamin B3, and CAVAQ10 could exert protective effects, in hypothalamic cells, through distinct mechanisms of action.

The greater efficacy of the fixed combination over the single components could depend on the fact that each molecule exerts, at least in part, their activity on mitochondria. As stated above, coenzyme Q10 functions as an electron accepter from mitochondrial complexes I and II, thus increasing the energetic rate of cells [17]. On the other hand, citicoline, performs a main function which is to maintain proper levels of cardiolipin and sphingomyelin in the cellular and axon membranes, and to stimulate the production of cardiolipin also within the mitochondrial membranes [5]. At this level cardiolipin is essential for the optimal activity of the enzyme complexes of the electron transport chain and, therefore, for ATP production. Williams et al., in one of the most interesting experimental studies on glaucoma, reported that the oral supplementation of vitamin B3 (nicotinamide or niacin, a precursor of nicotinamide adenine dinucleotide (NAD^+^)) or Nmnat1 gene therapy prevented retinal ganglion cells soma loss in aged mice. This was especially the case when administered in higher doses, and by worked inhibiting the formation of dysfunctional mitochondria with abnormal cristae, thus supporting mitochondrial health and metabolism [22].

It has been widely demonstrated that several molecular mechanisms were involved in the pathogenesis of NDs and that among all potential mechanisms, mitochondrial dysfunction represents a crucial step. Moreover, the mitochondrial impairment was also strongly involved in maintaining and fostering the neurodegenerative process [45,46].

Nowadays, besides aging, oxidative stress, apoptosis, and neuroinflammation, the mitochondrial dysfunction is becoming an emerging and crucial pathological process in the etiology of several neuro-ophthalmic disorders. The mitochondrial dysfunction results in depletion of ATP, halting the activities of the enzymes of the electron transport chain, the generation of reactive oxygen species, the reduction in mitochondrial DNA, and caspase 3 release. Thus, the concomitant administration of citicoline, CAVAQ10 and vitamin B3 has the theoretical advantage of better limiting the mitochondrial dysfunction, since the fixed combination could intercept different patho-physiological moments of the mitochondriopathy, such as oxidative stress, inflammation, apoptosis, and impaired cellular metabolism.

This study presents some important limitations. A partial limitation is that we did not perform the experiments on retinal ganglion cells or visual cortex neuronal cells, which are the cellular target of the most common forms of NDs of the visual system. This was due to the difficulties in obtaining retinal ganglion cells for cultures in our country. However, as stated above, hypothalamic cells were previously used to explore antioxidant and neuroprotective effects [7,38]. Nevertheless, further studies on retinal ganglion cells are required to confirm these preliminary results. Additionally, although Vitamin B3 and citicoline and coenzyme Q10 are widely used in different clinical conditions, we cannot definitively speculate about the possible utility of this association in treating NDs, since we performed an in vitro study on a neuronal cell model. Therefore, further dietary intervention studies in animals and humans are warranted to clarify whether these preliminary results can be really translated in clinical practice.

## 5. Conclusions

In conclusion, the present study showed different potentially positive effects of citicoline, vitamin B3, and CAVAQ10^®^ on neurons exposed to oxidative stress, although with varying potency and efficacy of each single component, and with greater effects of the fixed combination.

From a speculative point of view, we can make the following considerations.

First, because the three compounds exert at least in part their activity on the mitochondrion, they could induce synergistic effects on these crucial cellular organelles, leading to a significant metabolic burst for neuronal cells. Considering the role that mitochondria have in the pathogenesis of several NDs, this aspect warrants particular attention.

Second, because neuro-inflammation is increasingly advocated as one of the main contributing factors for the neuronal damage in several NDs, the effects of these three compounds on the reduction in markers of inflammation may potentially open new treatments modalities to reduce the damage progression.

Third, the antioxidant properties, along with the capability to increase levels of some neurotrophins, give this association potential neuroprotective properties.

Therefore, because these compounds exert their effects in different modalities, and because the most part of NDs are multifactorial, the association of citicoline/vitamin B3/Coenzyme Q10 could theoretically offer a multifactorial approach to contain neuronal cell damage or neural loss.

If these preliminary in vitro results were to be confirmed in dietary intervention studies in humans, they could open the possibility of evaluating the neuroprotective effects of the citicoline/vitamin B3/CAVAQ10^®^ fixed combination in some of the most important sight threatening diseases, such as glaucoma and diabetic retinopathy.

## Figures and Tables

**Figure 1 nutrients-14-02963-f001:**
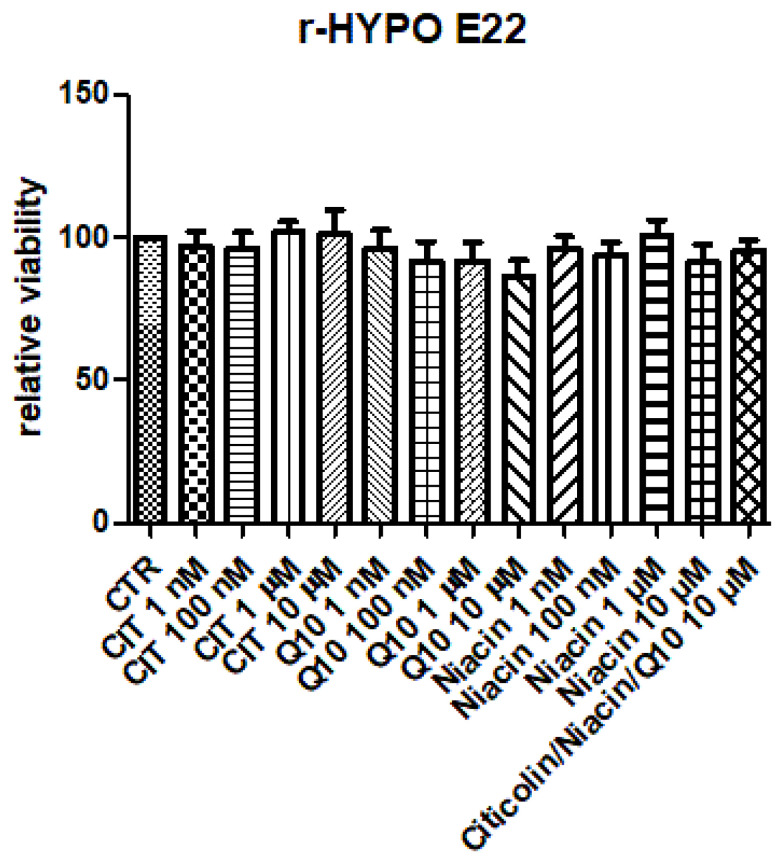
Null effect of citicoline, niacin (vitamin B3), coenzyme Q10 (1 nM–10 µM), and their fixed combination (10 µM) on hypothalamic HypoE22 viability. The cell viability was measured via MTT test. The null effects induced by single ingredients and their association indicated a complete biocompatibility in the selected in vitro model.

**Figure 2 nutrients-14-02963-f002:**
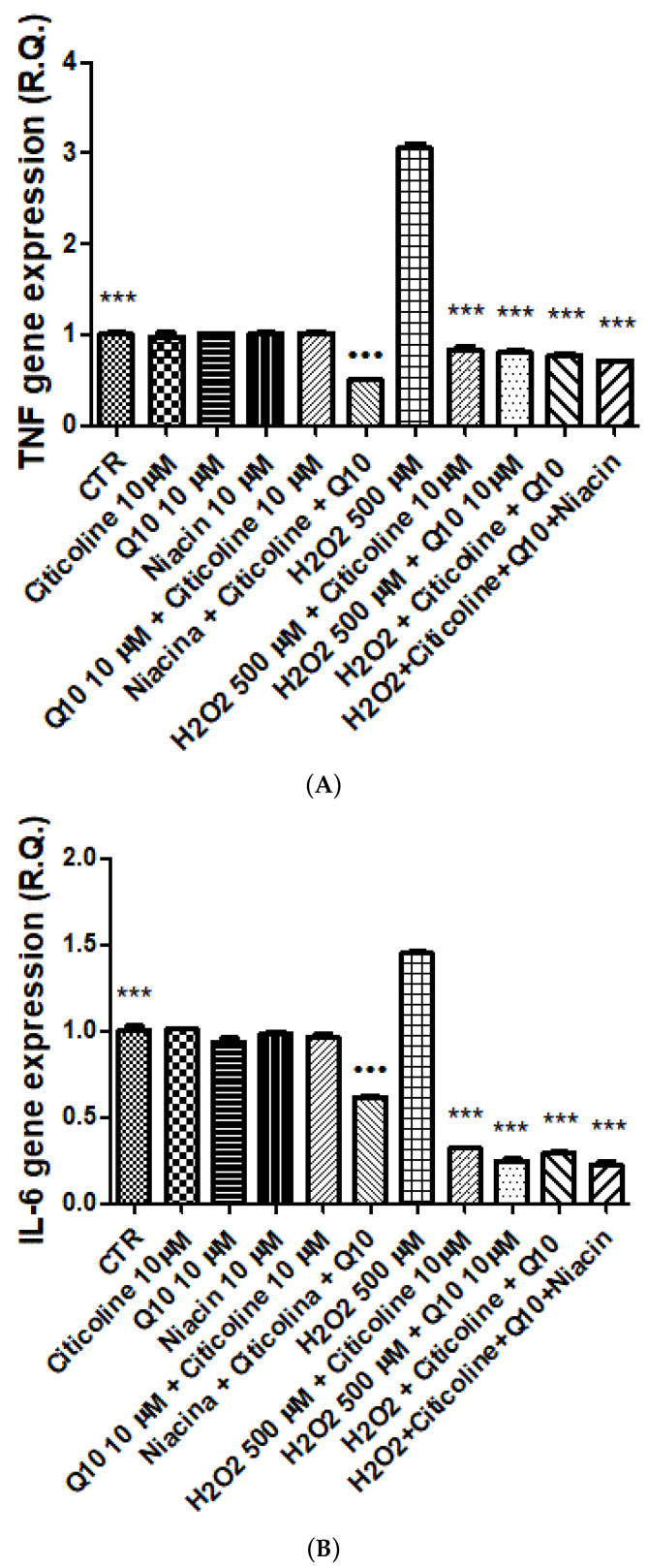
Effects induced by citicoline, niacin (vitamin B3), and coenzyme Q10 and the fixed combination on tumor necrosis factor-α (TNFα) (**A**), interleukin-6 (IL-6) (**B**), and brain-derived neurotrophic factor (BDNF) (**C**) gene expression, in isolated hypothalamic specimens exposed to hydrogen peroxide. ANOVA, *p* < 0.0001; *** *p* < 0.001 vs. control (CTR) group, **^…^**
*p* <0.001 vs. H_2_O_2_. The concomitant stimulation of BDNF gene expression and the reduced gene expression of both IL-6 and TNFα indicate neuroprotective properties by the single ingredients and their association. The latter is more effective in stimulating BDNF gene expression, compared with the single ingredients.

**Figure 3 nutrients-14-02963-f003:**
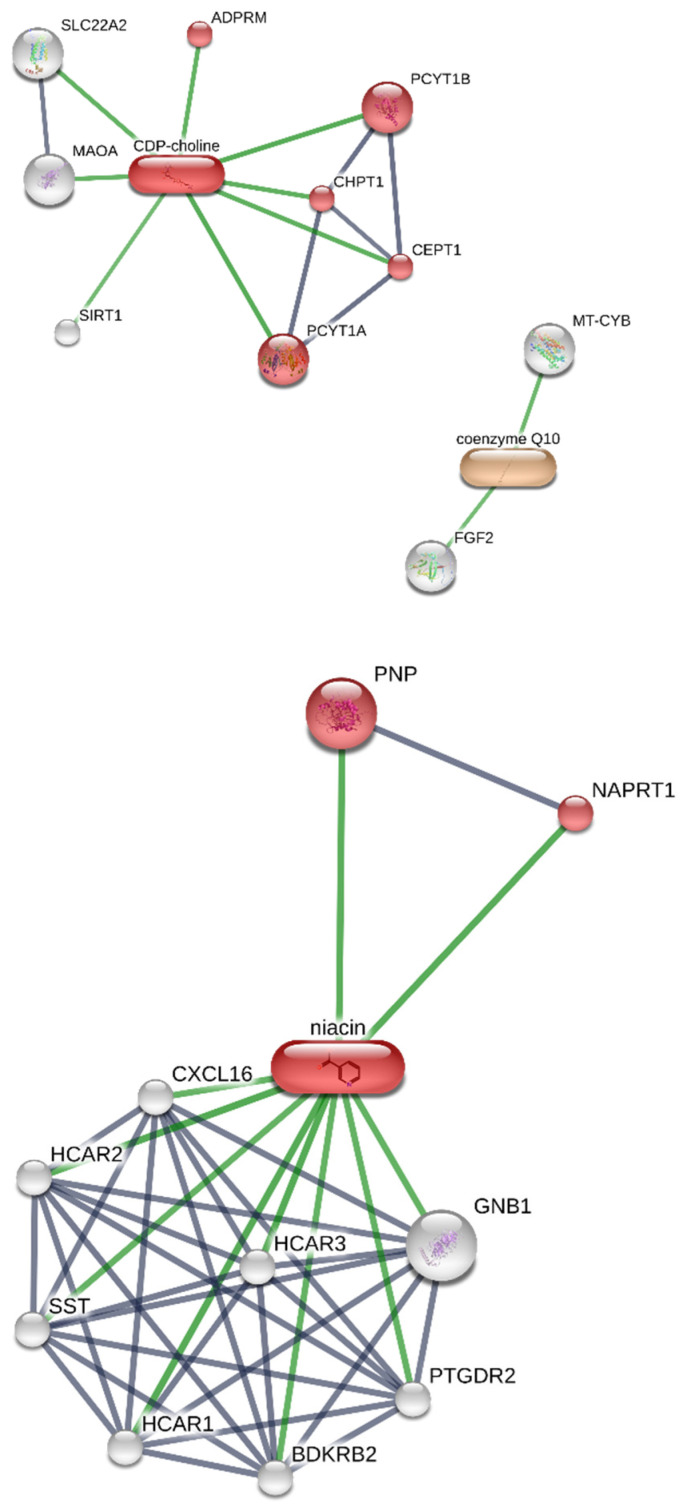
Results of the components-targets analysis conducted via the platform STITCH. The results suggested the capability of citicoline to interact with monoamine oxidase A (MAO-A). Additionally, citicoline was predicted to modulate dopamine uptake, through interaction with the neurotransmitter carrier (SLC22A2), and through phosphatidylcholine synthesis, in which it interacts with phosphate cytidylyltransferases (PCYT1B and PCYT1A). On the other hand, CAVAQ10 was predicted to interact with cytochrome b (MT-CYB), involved in the ATP biosynthesis, and fibroblast growth factor-2 (FGF2).

## Data Availability

The data presented in this study are available on request from the corresponding author.

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
