# Peer review of "Citicoline/Coenzyme Q10/Vitamin B3 Fixed Combination Exerts Synergistic Protective Effects on Neuronal Cells Exposed to Oxidative Stress"

_nutrients, 2022, doi:10.3390/nu14142963_

Round 1
Reviewer 1 Report
Authors addressed all the concerns that I had in the past. The manuscript can be accepted.
Author Response
Dear Reviewer, we thank you the constructive comments you previously provided, and we would like to thank you for the appreciation of our work.

Reviewer 2 Report
The nutrients-1794777 tried to address the antioxidative effects of the vitamin mixture. However, this was not a dietary intervention; therefore, applicability to the potential clinical use is not ensured. Moreover, the authors solely relied on the qRT-PCR data (it should be merged as one figure); therefore, the biological significance is not warranted.
1. Manuscript does not develop the scientific logic detail. 2. Poorly written English. 3. Figure captions are not written with detailed information. 4. Journal format should be followed. (font type, size, references, and so on).
Author Response
The nutrients-1794777 tried to address the antioxidative effects of the vitamin mixture. However, this was not a dietary intervention; therefore, applicability to the potential clinical use is not ensured.
Thank you for the correct comment. We agree with you that this was not a dietary intervention and, thus, any clinical conclusion is not pertinent. However, since these compounds are broadly used in the management of some neuro-degenerative conditions in humans, we tried to make some speculations.
In accordance with your suggestion, we better clarified this aspect modifying the discussion in different sections and expanding the limitation section.
Moreover, the authors solely relied on the qRT-PCR data (it should be merged as one figure); therefore, the biological significance is not warranted.
Thank you for your comment. Figure has been merged in one figure, which is now Figure 3 A, B, C. The consecutive number of figures has been updated.
- Manuscript does not develop the scientific logic detail.
The corrections on RT-PCR data have been done.
- Poorly written English.
As you kindly required, a native English speaker edited the manuscript
- Figure captions are not written with detailed information.
As you kindly required, figure captions have been revised.
- Journal format should be followed (font type, size, references, and so on).
In accordance with your recommendation, we revised the text in accordance with the Journal style guidelines.

Reviewer 3 Report
This is a manuscript about an interesting topic, the possible antioxidant and anti-inflammatory effects on neuronal cells caused by the combination of citicoline, vitamin B3, and Coenzyme Q10. It is also well structured.
The introduction section gives important information about the background of the study.
The results are well-presented and adequately analyzed.
The discussion section might need to be improved. The authors need to highlight in a better way the advantages and disadvantages of their study.
English language and style are generally fine.
The conclusions section could be written in a more critical way.
Author Response
- This is a manuscript about an interesting topic, the possible antioxidant and anti-inflammatory effects on neuronal cells caused by the combination of citicoline, vitamin B3, and Coenzyme Q10. It is also well structured.
- The introduction section gives important information about the background of the study. The results are well-presented and adequately analyzed.
Dear Reviewer, we thank you for your appreciation.
- The discussion section might need to be improved. The authors need to highlight in a better way the advantages and disadvantages of their study.
As you kindly required, we extended the discussion section, stating what could be the pros & cons provided by the results of this study. The limitation section was accordingly implemented.
- English language and style are generally fine.
A mild English editing has been done.
- The conclusions section could be written in a more critical way.
As you kindly recommended, we modified the concluding paragraph, giving it a more critical interpretation.

Round 2
Reviewer 2 Report
The nutrients-1794777 is updated and may be acceptable.